Publication trends in neuroimaging of minimally conscious states

Garnett Alex 1
Lee Grace 2
Illes Judy 2 jilles@mail.ubc.ca
1 Simon Fraser University Library , Burnaby, BC , Canada
2 National Core for Neuroethics, Division of Neurology, Department of Medicine, University of British Columbia , Vancouver, BC , Canada
Gelfand Mikhail
Electronic publication date: 2013 Sep 19
Publication date: 2013
Volume: 1
Electronic Location ID: e155
Received 2013 Jun 19; Accepted 2013 Aug 18
Copyright: © 2013 Garnett et al.
Copyright year: 2013
Copyright holder: Garnett et al.
License: This is an open access article distributed under the terms of the Creative Commons Attribution License, which permits unrestricted use, distribution, and reproduction in any medium, provided the original author and source are credited.
License URL: https://creativecommons.org/licenses/by/3.0/

Keywords: Minimally conscious states, Persistent vegetative state, Clinical research, Neuroimaging, Bioethics, Altmetrics

Funding: CIHR EOG #120257 CNE #85117 The National Core for Neuroethics received grants from the Canadian Institutes of Health Research: EOG #120257 and CNE #85117. Judy Illes is the Canada Research Chair in Neuroethics. The funders had no role in study design, data collection and analysis, decision to publish, or preparation of the manuscript.

==============================
We used existing and customized bibliometric and scientometric methods to analyze publication trends in neuroimaging research of minimally conscious states and describe the domain in terms of its geographic, contributor, and content features. We considered publication rates for the years 2002–2011, author interconnections, the rate at which new authors are added, and the domains that inform the work of author contributors. We also provided a content analysis of clinical and ethical themes within the relevant literature. We found a 27% growth in the number of papers over the period of study, professional diversity among a wide range of peripheral author contributors but only few authors who dominate the field, and few new technical paradigms and clinical themes that would fundamentally expand the landscape. The results inform both the science of consciousness as well as parallel ethics and policy studies of the potential for translational challenges of neuroimaging in research and health care of people with disordered states of consciousness.

Introduction

Bibliometric and scientometric methods provide a means of charting trends within scholarly publications (Borgman, 1990). Beyond simply establishing the rate of publication of a given author or topic and counting citations, they can be used to chart the establishment of emerging fields within broader disciplines. In this study, we apply existing and new bibliometric methods to neuroimaging research of minimally conscious states (henceforth NiMCS), a domain of research that has its roots in the early 2000s and that has potentially important implications for the health care of people with traumatic brain injuries (TBIs).

NiMCS is directly concerned with neuroimaging for patients with disorders of consciousness – a set of conditions that span the fully unresponsive vegetative state (VS) to the more intermittently responsive, minimally conscious state (MCS). Acquired and traumatic brain injuries that can give rise to disorders of consciousness account for an estimated 57 million people worldwide (Langlois, Rutland-Brown & Wald, 2006). In the United States, an average of 1.4 million TBIs occur each year (Langlois, Rutland-Brown & Wald, 2006; Roozenbeek, Maas & Menon, 2013).

Recent advances in research using functional neuroimaging technology have provided novel methods to assess consciousness in those patients who remain impaired, and offer the potential of improving diagnosis and possibly a means of communicating with them through measures of brain activity (Fernández-Espejo et al., 2011; Monti et al., 2010; Owen & Coleman, 2008; Owen et al., 2006). Functional magnetic resonance imaging (fMRI), for example, measures changes in regional cerebral blood flow and yields activation maps of cognitive processes (Boly et al., 2008; Schiff et al., 2005; Monti et al., 2010). The use of fMRI is limited, however, given scanner cost, access, and the still evolving science especially as it pertains to the interpretability of the signals. A historically older method that measures electrical activity from the scalp – electroencephalography (EEG) – has also been tested for this application, with success in revealing differences between levels of disordered consciousness (Gosseries et al., 2011; Laureys et al., 2010). While information about the speed of processing from EEG exceeds that of fMRI, information from EEG about the spatial distribution of function across brain regions is comparatively incomplete. Nonetheless, EEG does offer important advantages including lower costs for purchase and maintenance, and portability for use at the bedside. Eventually, a combination of both technologies may be the solution of choice in the translational trajectory.

The enthusiasm for the basic science of consciousness and promise of clinical utility in improving the diagnosis and prognosis of people with TBI with either single or multiple imaging modalities have been accompanied, if not fuelled by the press. Coverage of promising innovation is an opportunity for public education, but headlines such as “Vegetative patient ‘speaks’ to doctors through scanner” (BBC News November 2012) and “People appear to dream while in minimally conscious state” (US News August 2011), can lead to misunderstanding (Racine, Bar-Ilan & Illes, 2005; Racine, Bar-Ilan & Illes, 2006). Moreover, media visibility, taken together with both medical significance and highly publicized internal debates about technical details of study design and data analysis (Goldfine et al., 2013; Cruse et al., 2011; Cruse et al., 2012; Cruse et al., 2013), leave the Academy and public alike wondering about what is true about the evolution of the domain and, by extension, what the evolution signals for the future of the domain.

To address these challenges, we turned to bibliometric analysis to derive a top-down, topical review of the research (Mörchen et al., 2008; Small, 2006). We posed the following specific research questions: (1) What are the publication growth patterns in NiMCS as reflected in the peer-reviewed literature?

(2) Who is contributing to research in this field?

(3) How are authors in this field related to one another?

(4) What are the key translational issues?

To answer these questions, we used existing methods as well as customized innovative methods to map publication trends and the degree to which the relevant literature gains new authors. We also incorporated qualitative analysis of MeSH terms to understand the fields in which authors have published, and carried out a clustering of articles with a visualization component for comparisons among subfields (Struble & Dharmanolla, 2004; Yamamoto & Takagi, 2007). With these data, we explore the historical path of NiMCS research and discuss the future of the research as it can be expected to further unfold.

Methods

Publication patterns

To calculate basic publication trends in NiMCS, we created the following PubMed query using the Medical Subject Headings (MeSH) keyword vocabulary: “[minimally conscious states] and ([neuroimaging] or [magnetic resonance imaging] or [functional neuroimaging])”. This retrieved all indexed articles within PubMed matching the subject headings,3 including primary research as well as reviews and editorials. We also documented the journals in which they were published (neuroscience, bioethics, general science). We determined the mean growth rate of NiMCS publications per year in the time period between 2002 and 2011; data from 2012 was not included as it was too recent to provide reliable metrics (Bornmann, 2013). Because our analysis covered a relatively short period, we also determined the change in growth rate each year, in order to determine whether this rate changed meaningfully within the timeframe. In addition to the base publication rate, we calculated the mean number of authors per publication in each year over the time period of interest. For comparison, we calculated the mean growth rate in all biomedical literature, defined for the purpose of this study as all papers indexed in PubMed over the same time period.

Author characteristics

To determine the rate at which NiMCS gains new researchers, we calculated the number of unique authors publishing each year who had never previously published within the discipline, as well as the number of unique authors publishing each year who had not published within the discipline in the prior three years.

To determine the originating specialty of NiMCS authors, we created a script that utilizes PubMed’s Entrez API to aggregate the MeSH terms applied to the authors’ entire respective body of work as represented in PubMed (Cheung, 2012). We then calculated the variance in the application of MeSH terms to each of these authors’ respective bodies of work. We contrasted the variance in the application of MeSH terms with the total number of MeSH terms, and total number of unique MeSH terms applied to each data set to yield a measure of domain breadth versus total publishing output for all authors contributing to the NiMCS literature. Then, to provide individual examples of contributing authors from various domains, we randomly selected 10% of these authors for manual analysis of subject specialities as expressed by MeSH.

Author interconnections

We used the ISI Web of Science database and the CiteSpace and Gephi graph visualization packages to provide a measure of the interconnections of authors retrieved in the full database. We translated our PubMed query to the ISI equivalent terms (““minimally conscious” and neuro*”), retrieved matching articles,4 and exported the results to CiteSpace to generate a meaningful and interpretable graph. An automatic pruning step was performed in CiteSpace to prioritize strongly connected nodes.5 The data were then exported to the GraphML format used by the Gephi graphing software for visualization. Except where otherwise noted, the ForceAtlas2 graph layout algorithm was used (to improve readability by emphasizing nodes’ relationship with their “neighbours”), with color-coded clustering performed using Louvain Modularity (designed to find communities within ad-hoc networks; Blondel et al., 2008) and node weighting by eigenvector centrality. Two separate graphs were generated: (1) A map of author collaborations in which the nodes are individual researchers and the edges represent co-authorship.

(2) A map of article usage over time in which the nodes are published works and the edges represent citation. The top 10% of articles by node centrality from this graph were reviewed for content, and the graph was manually annotated with subtopic descriptions corresponding to groups of articles. Graphs were then interpreted qualitatively using visual information from the modularity and centrality weightings.

Clinical focus

We used the subset of Open Access articles from the total set retrieved for full-text analysis and coded the constituent papers for three factors: (1) Average number of brain injured subjects with a given diagnosis. We created a schema to organize the different descriptions used to classify patients (e.g., traumatic injury and non-traumatic injury, minimally conscious, and persistent vegetative state),

(2) Nature and extent of discussion of clinical implications, including economic impact. Each new theme was counted once per paper.

(3) Paradigms and stimuli used in the research protocols.

Results

Publication patterns

We retrieved 311 NiMCS papers for analysis, of which 141 (45%) were primary research articles. Eighty one (26%) represented case studies. Sixty-one (20%) were review articles, and the remaining 28 (9%) editorials and comments. The overall number of these NiMCS publications between 2002 and 2011 increased at an average rate of 27% per year (Fig. 1). However, the growth rate was not linear across this time period, increasing by over 100% between 2002 and 2005, then falling to an average rate of 7% per year between 2005 and 2011. When shifting this comparison by one year, the growth rate between 2006 and 2011 was 11%. By comparison, the growth rate of all biomedical literature (defined for the purpose of this study as all papers indexed in PubMed) over the time period of 2005 to 2011 was 7%. Using 2005 as the benchmark, the NiMCS growth since is essentially flat when normalized to the more broadly defined rate of scientific “inflation”.

Figure 1 Growth rate of neuroimaging and MCS publications by year.

The average number of authors per NiMCS papers increased at an average rate of 33% per year from 1.3 in 2002 to 5.5 in 2011, linearly across the time period. The number of authors publishing in NiMCS who had never published in the domain before decreased from 92% in 2003 (when most researchers were “new”) to 67% in 2011. Therefore, two-thirds of all authors currently publishing in NiMCS are likely first time authors. Meanwhile, the number of authors publishing in NiMCS who had not published in the domain during the preceding three years held fairly consistent at 80 ± 3% since 2007 (Table 1).

Table 1 Change in NiMCS publication and authorship rates from 2002–2011.

%New-Last refers to authors who are new to the discipline relative to the prior year; %New-Total refers to authors who had not published within the discipline for the prior 3 years.

Year	Number of	Ratio	%New	
	Authors	Publications		Last	Total	
2002	13	10	1.3			
2003	12	9	1.3	92%		
2004	30	12	2.5	97%		
2005	109	25	4.4	95%		
2006	29	22	1.3	72%	72%	
2007	77	25	3.1	81%	70%	
2008	87	27	3.2	83%	68%	
2009	173	42	4.1	82%	66%	
2010	191	41	4.6	79%	70%	
2011	176	32	5.5	77%	67%	

Author characteristics

Despite being a narrowly defined subfield within clinical neuroscience, NiMCS authors come from a very diverse set of background subject areas according to PubMed’s MeSH vocabulary. Few MeSH terms, among them “self-help devices” and “decision-making”, are common to multiple authors. Other contributing specialties include “Quadriplegia, Nerve Growth, Alzheimer Disease” (MA Bruno), “Sleep, Pain” (M Boly), “Neurophils, Anesthesia” (M Lamy), “Communication Aids, Behavioral Therapy” (MF O’Reilly), and “Sleep, Memory” (M Schabus). The variance in assignment of MeSH terms to each author’s work is provided in Supplemental Information, and the raw data containing term counts for each author is available upon request.

Author interconnections

Co-authorship graph computations illustrate that the most prominent author (eigenvector centrality = 1) largely publishes with members of his own research team. Similar separate author clusters are formed for two other prominent authors with eigenvector centrality = 0.032 and 0.042 and their within group co-authors. These and other high-degree nodes bridging multiple clusters are labelled on Fig. 2. The distribution of authors’ centrality values is long-tailed, with only two authors out of the top 58 (3.4%) having centrality greater than 0.5 and only six (10.3%) having centrality greater than 0.25.

Figure 2 Co-authorship graph of NiMCS and related research.

Nodes represent authors; edges represent co-authorship. Graph layout uses the ForceAtlas2 algorithm. Clusters are calculated via Louvain modularity and delineated by color. Frequency of co-authorship is calculated via Eigenvector centrality and represented by size.

The co-citation graph reveals dependencies between disciplines in NiMCS (Fig. 3). The lower portion of Fig. 3 shows another kind of “long tail” of straightforwardly empirical research into unconscious stimuli and some generalized studies of brain function using positron emission tomography (PET). These studies inform bioethical and philosophical work on the neuroscience of consciousness. This bioethics literature is most closely related to clinical neuroscience in that it focuses specifically on differentiating PVS and MCS, and on issues around communication with patients by measuring functional brain activity. The most frequently cited papers (corresponding to increasing node size) in this domain were primarily published between 2000 and 2005, consistent with the levelling off of the publication growth rate.

Figure 3 Co-citation graph of NiMCS and related research.

Nodes represent papers; edges represent citation. Graph layout uses the ForceAtlas2 algorithm. Clusters are calculated via Louvain modularity and delineated by color. Citations are calculated via Eigenvector centrality and represented by size. Subtopic labelling is performed via manual consideration of the articles.

Clinical focus of the research

In the open access subset of 32 papers, different levels of granularity were used to describe levels of consciousness of patient-participants. Some publications dissociated MCS and PVS clearly, while others combined patients into a single “non-communicative” group, and contrasted them only with healthy control subjects. We created the schema displayed in Fig. 4 to manage the different descriptions for the purpose of this comparison.

Figure 4 Schema for grouping of diagnosis categories derived from manual consideration from NiMCS literature open access sample.

Articles reported approximately equal numbers of PVS and MCS subjects, with a majority of MCS subjects categorized as having a traumatic injury. The mean number of non-communicative subjects per study was 9, while the mean number of healthy subjects was 18.

Of the 32 publications analyzed qualitatively, 15 were primary research reports. Of the primary research articles, 8 (53%) featured some form of clinical discussion; 100% of publications not reporting primary research contained discussion of clinical issues. Clinical assessment was the dominant theme across all types of papers studied (N = 13/32). We also noted specific discussions of recovery of consciousness (N = 6/32), ethical decision-making in patient care (N = 5/32), and clinical management by the bedside (N = 2/3). Three non-primary research articles discussed economic implications. For example:

…[we] believe costs should not figure as a priority in an ethical discussion, and we do not believe that indefinite continuation of life-support is the only ethical option.

Panksepp et al., 2007. Bioethics review article

Of the papers reporting stimuli used as part of a human subjects experiment, 8 described resting state activations; 7 described experiments involving sensorimotor and verbal stimuli, and spatial navigation.

Discussion

We applied sophisticated bibliometric and scientometric methods to NiMCS publications to characterize the growth of the NiMCS field in terms of output, author characteristics and origins, and focus. In light of the possibility that advances in this domain might well revolutionize the health care of people with acquired brain injuries and disorders of consciousness (Fins et al., 2008), and the acute media attention given to steps along the research trajectory, we were motivated to map the academic landscape of the field and provide an empirical perspective on its trajectory. We used both existing tools (Chen, Ibekwe-SanJuan & Hou, 2010) and customized methods to achieve this goal.

It is important to note that bibliometrics, while a powerful and well-researched means of exploring the published scholarly record, are imperfect for representing informal contributions to science (i.e., not indexed by PubMed and/or Thomson-Reuters) or those which have been published very recently due to the procedural lag in citations. These limitations are not specific to our research, and we have attempted to address them by not including data newer than 2011. At the same time, we deliberately did not try to normalize citation totals of articles for the time since their publication; just as we did not explicitly rank articles using these data, we sought to obtain historical data pertaining to the development of NiMCS research to date. Additionally, we did not perform any manual disambiguation of author names. When our data source failed to resolve the difference between “John Smith” and “JO Smith” we made no effort to correct this,6 thus some prolific authors may not have all of their published material assigned to the same name string, and their ranking in our graph may be affected insignificantly. Additionally, the findings from our full-text content analysis may be biased toward more recent publications and may not be entirely representative of the subject domain for some authors due to our decision to include only open access research. However, given that much of the research under consideration falls roughly within the time period that the US National Institutes of Health PubMed Central deposit mandate has been in effect, we do not believe that it has significantly affected our results, and include this statement to acknowledge that our selection was not strictly random. Finally, because this was an exploratory study describing the growth of a new field, we do not have adequate statistical baselines for much of the work we have undertaken, and cannot make many comparisons.

With these limitations in mind, we found that the number of NiMCS publications per year is increasing, and that there are conventional, upward trends for the number of authors per paper and for the rate at which new authors are added (the “replacement rate”). These are consistent with the establishment and subsequent formalization of biomedical subdisciplines. The content analysis reinforces the fact that NiMCS is, in large part, a clinical research endeavour. There is an unusually large amount of clinical and ethical discussion compared to other similar analyses of primary research in neuroscience (Garnett et al., 2011). Descriptions of the consciousness of patients are highly variable. Like the pool of stimuli that is small and unchanging, subject numbers are also low and steady per study. The highly focused co-authorship patterns suggest that NiMCS has not yet become widespread as a subdomain within clinical neuroscience. Despite the substantial author replacement rate, there appears to be only a small cohort of thought leaders and few new themes beyond clinical assessment that would fundamentally expand its landscape.

Of the findings from our content analysis, many – such as the limited clinical and ethical discussion in primary research, and the number of subjects per study – were also largely confirmatory. This does not diminish their value, particularly when taken together with some of our author-level metrics. Notice that S. Laureys is the most central node in the co-authorship graph; he is also the leader of the Coma Science Group in the Department of Neurology at the Liege University Hospital in Belgium, and the cluster of co-authors surrounding him in the graph is comprised largely of his own graduate students. This is consistent with Abbasi, Altmann & Hossain (2011), who determined that working with many students is generally a more straightforward way to accumulate citations – academic capital – than working with other well-performing scholars. NiMCS research is also necessarily constrained in terms of actual capital, dependent on a teaching hospital to accommodate its teaching component. Thus, one should not necessarily expect a large cohort of thought leaders. This is particularly true when the existing thought leaders are distributed across international borders, which generally results in a diminished citation effect from collaboration (Abbasi & Jaafari, 2013). However, there is no reason to believe that this is unique to NiMCS research. Interdisciplinarity in science is increasing generally (Silva et al., 2013) and, in this and other disciplines where it is functionally difficult to contribute at the perceived top-end of the scientific dialogue, it is more productive – at least from a bibliometric perspective – to look at more diverse contributions as we have done.

Will the science of consciousness realize its hope to change patient management after brain trauma as some have debated (Fins et al., 2008; Owen et al., 2006; Owen & Coleman, 2008; Laureys et al., 2010)? Time will tell, and further analyses such as those conducted here will provide the lens for continued inquiry. But, even if the domain cannot deliver on its health care goal given potentially insurmountable complexities of consciousness, meaningfulness of communication, scarce resources and other variables, there is tremendous knowledge to be gained. We are learning deeply about the meaning of signals from the brain in health and in diseases of the central nervous system, about the boundaries and limits of this neuroscience research, and about the importance of balanced communication in the dissemination of information and neuroliteracy among the public.

Supplemental Information

Supplemental Information Variance in application of MeSH terms to NiMCS authors

Click here for additional data file.

Additional Information and Declarations

Competing Interests

Author Contributions

3 PubMed convention is such that formatting a search in this way will search both matching MeSH headings and matching free-text strings, in order to broaden coverage.

4 We verified that the ISI query result set was not more than 20% different from the PubMed query result, determined to be an acceptable margin of error for cross-database comparison.

5 This pruning step was performed specifically to compress the graph into a viable 1000px horizontal image, which can be easily scanned by the human eye. Data were pruned using a percentage cutoff (15%) for the most-cited articles within the dataset for a given year, rather than pruning the entire dataset, in order to surface “smaller” contributions on a year-by-year basis and provide a counter to the Matthew Effect (Merton, 1968) and then compiled. This ceiling was adjusted multiple times after consultation with domain experts in order to ensure that it adequately represented the social structure of the domain. The unpruned data are available as a raw node-and-edge graph file upon request for readers who wish to have a 100% complete representation of co-authorship or co-citation in NiMCS.

6 Until proposed solutions to this issue such as http://orcid.org/ enjoy wider uptake, we believe that it is methodologically cleaner not to attempt to resolve ambiguous author names manually.

The authors have no competing interests.

Alex Garnett conceived and designed the experiments, performed the experiments, analyzed the data, contributed reagents/materials/analysis tools, wrote the paper.

Grace Lee and Judy Illes analyzed the data, wrote the paper.

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
