# Peer review of "Publication trends in neuroimaging of minimally conscious states"

_PeerJ, doi:10.7717/peerj.155_

## Round 0.1 · original submission · Major Revisions

In addition to the reviewers' comments, the following points should be taken into account.
One major issue not addressed in the text is disambiguation of homonymous author names. Chimeric “authors” assigned with papers (MeSH terms etc) by different people may (or may not) severely confound the analysis. This needs to be addressed or at least discussed.
Statements such as “(something is) climbing linearly” need to be supplied with a calculation of statistical significance or re-phrased.
The Methods session is not sufficiently detailed, contains many implicit assumptions and undefined terms, does not provide motivation for specific analyses. E.g. why calculation of node centrality (ca. line 98) is relevant in the context of the study? why pruning is sufficient to counteract the Matthew effect (footnote note 1 to line 90)?
The Introduction lacks references to many relevant papers on similar bibliometric studies. In fact, the manuscript shows almost no connection to previous work.
The term “low-level research” (line 150) sounds denigrating, although probably not intentionally.
How is correction for years since publication applied (if any)? – this is relevant when, e.g., most frequently cited papers are discussed (line 155).
Trends for NiMCS papers are not compared with trends in similar or generic areas: hence, no comparison can be made and no conclusions formed.

·

Basic reporting

- In order to aid the reader in understanding the full scope of the article please provide a short description of the techniqual terms such as 'ForceAtlas2 algorithm' and Louvain modularity.

Experimental design

- Do the authors have also searched PubMed in 'free text' mode? This is important as not all relevant/recent articles are MeSh indexed.
- Choosing to include only open acces articles may introduce a bias as the 'top gear'
journals are unfortunately usually not open access. However, I understand very well
that this method is likely dictated by practical considerations. The authors should
therefore at least mention this point in the discussion section of the article.
- The authors note that in the field of NiMCH the average number of authors is gradually increasing. Although this is true, it is not an exception as the average
number of authors on biomedical papers in general is gradually increasing as more
and more researchers begin to collaborate.

Validity of the findings

- The authors should include a paragraph in the discussion section of their manuscript elaborating on the potential limitations of their method. In particular it is important to stress that bibliometric analysis is only a very indirect measure of the impact and direction of a field, for example, as it is necessarily based on past results, In other words, the most recent ideas, conceps and results are simply waiting to be published.

Reviewer 2 ·

Basic reporting

Excellent-- a scholarly presentation

Experimental design

Study used appropriate, and widely accepted bibliometric methods in their design

Validity of the findings

Findings are rather straightforward and corroborate prior research

Additional comments

I wish that the author(s) had expanded in their discussion , in more detail, regarding the specific topical areas in Neuro-Imaging or MCS that were highly ranked in the analysis. The reasons why some topics are emphasized (& others not) should be explored.

---

## Round 0.2 · Minor Revisions

The revised manuscript still does not address the issue of statistical significance.
Editorial: the sentence “While the information about the speed of processing that EEG exceeds that of fMRI, information about the spatial distribution of function across brain regions pales comparatively” seems to be incomplete.
Unfortunately, the revised manuscript did not contain tracked changes, as it has been suggested in the decision letter.

---

## Round 0.3 · accepted · Accept

While I still think that some estimate of statistical significance could improve the paper, I do not insist.